# Sexuality and Affection in the Time of Technological Innovation: Artificial Partners in the Japanese Context

**Beatriz Yumi Aoki** [1,*] and **Takeshi Kimura** [2]

1    Communication and Semiotics Program, Pontifícia Universidade Católica de São Paulo, São Paulo 05014-901, Brazil

2    Faculty of Humanities and Social Sciences, University of Tsukuba, Ibaraki 305-8571, Japan; takeshi-kimura.fw@u.tsukuba.ac.jp

\*    Correspondence: beatrizyaoki@gmail.com

**Abstract:** Recent years have witnessed an increase in the number of academic studies on the impact of technological advancements on human life, including possible transformations and changes in human sexuality following the development of sex-related devices, such as sex robots. In this context, terms such as posthuman sexuality, digisexuality, and techno-sexuality have emerged, signaling possible new understandings of sexual, intimacy, and emotional practices. It is important to note that ancient history shows that humankind has for a long time been fascinated with their relationship to non-living things, mostly human-like figures, such as dolls. The Ningyo (人形, the Japanese term for doll) has a long history of usage, and has deep religious and animistic significance in the Japanese context—there are records of sexual use as early as the 18th century. With this context in mind, this paper focuses on three Japanese examples, aiming to shine a light on beyond-human relationships, which include a Japanese man's marriage to a digital character, sex dolls, and communicative robots, from both a sexual and emotional perspective. In a new horizon of sexual and romantic possibilities, how will humans respond, and what can emerge from these interactions?

**Keywords:** digital character; sex dolls; sex robots; spirituality; Japanese sexual culture



## 1. Introduction

Technological developments have been strong forces to help transform social relationships and human behavior, including sexual practices. For example, IT and AI have helped global societies to deal with, and adjust to, lockdown as a social device, while global communities have been fighting the COVID-19 pandemic since 2020. Among the more unexpected technologies to contribute to this adjustment were sex-tech devices—sex dolls and sex robots—which helped people maintain sexual wellness during the pandemic. During this period, there was an increase in their use and sales in many countries (Lee 2020; Owsianik 2020). Especially noteworthy were the higher sales of sex robots (Morris 2020). Matt McMullen, CEO of Abyss Creation—one of the leading companies in the US and worldwide market—affirms that their sex doll and sex robot sales increased about 75%, compared to the pre-lockdown period. Due to social distancing measures, dealing with social anxiety, loneliness, skin hunger (Park 2020)[1], and even boredom, as well as preserving sexual wellness through these devices, have become important topics of discussion[2]. In September 2020, a British TV show interviewed Jade Stanley, the owner of the company Sex Doll Official, who referred to the sex doll as a companion doll for those who are unable to have a human partner (This Morning 2020). It is worth highlighting that loneliness was recognized as a serious social issue in Britain even before COVID-19, as the Ministry of Loneliness was established in 2018 (Mencken 2020; Sang-Hun 2020).

Over the last few years, academic studies on sex dolls, sex robots, and sex-tech devices, as well as possible new understandings of human sexuality—such as the concepts of digisexuality and techno-sexuality—have increased in number and quality. The social

changes during the COVID-19 pandemic demand a more nuanced scholarly and social discussion by introducing more culturally and psychologically appropriate sensitivity into digisexuality and techno-sexuality. Since some studies on techno-animism (Jensen and Blok 2013) might be related to the concepts of digisexuality and techno-sexuality, a much broader and more inclusive perspective is required. Recently, there has been an increase in scholarly discussions of the relations between religion and robots (Trovato et al. 2019). Therefore, a broader and more inclusive notion of religion and spirituality is necessary because, in Western academic discussion of digisexuality and techno-sexuality—including sex dolls and sex robots—religion often means Christianity and other institutionalized forms of religions. Owing to the bias derived from a narrow definition of religion, scholarly discussion on the topic globally misses other important, more subtle, and nuanced aspects of its spiritual and religious dimensions by employing a broader definition of religion.

By locating digisexuality and techno-sexuality in the context of Japanese sexual, cultural and religious histories, this paper examines the emotional and sexual contours of the human–technology relationship by analyzing emerging conceptions of sexuality and affection in conjunction with technological innovations. The Japanese cultural history of sexuality—which draws both positive and negative reactions from global communities—is chosen for its complex and multifaceted aspects[3]. This paper argues that by unraveling notions of affection and digital intimacy, it is more understandable that by paying more attention to cultural and spiritual backgrounds, an emerging, technologically enhancing condition would enable people to explore and develop alternate ways of expressing affection and sexuality, blurring the boundaries between body and technology, and real and virtual.

The structure of this paper is divided into five sections, in addition to its Introduction. Section 2 presents a literature review on recent studies on sex dolls and robots, a relatively new area of inquiry. Section 3 examines three cases from contemporary Japan: a young Japanese man who married the virtual character Hatsune Miku, and the authors' partial research results from online surveys completed by both love doll owners and Robohon (a mobile phone in the shape of a little robot) owners. Section 4 briefly examines Japanese dolls, technology, and sexuality throughout history. Finally, Section 5 argues that by introducing a notion of homeostasis, a more nuanced and bidirectional affectionate and emotional connection would occur in the human–robot relationship.

## 2. Literature Review

Before reviewing scholarly studies on digisexuality and techno-sexuality, it is useful to discuss the current state of sex robot development, which is still far from producing a fully functional and idealized form of sex robot. Different types of sex robot are being developed and sold in countries such as the USA and China. In the USA, Harmony is one of the most prominent current representatives of its kind. Created and sold by Realbotix, Harmony is a joint venture between three organizations: Abyss Creations/Real Doll, a North American company which has produced sex dolls for about 20 years; Daxtron Labs, a company focused on AI and communications solutions; and NextOS, a mobile-agents provider. Harmony is an AI-driven robotic doll system "designed to be a customizable personal companion agent capable of interaction via the user's phone, a virtual reality (VR) headset, as well as through a physical interactive android body" (Coursey et al. 2019, p. 78), which allows for the definition of personality traits and is equipped with a conversational system. A Chinese AI-Tech company has also created sex robots called Emma and Feifei. On the company's website, they are called companion (or even girlfriend) robots rather than sex robots. They are equipped with a talking system (available in Chinese- and English-speaking versions), deep learning technology (enabling her to store the information she receives from interactions and learn from them), facial expressions, eye, mouth and neck movement, internal heating and sensors, and a moaning system. In terms of physical appearance, both robots follow the characteristics of and aim to resemble a human figure, which would characterize them as humanoid robots. By definition, the humanoid form

"is intended to represent (and is taken to represent) a human or human-like being in its appearance" (Danaher 2017, p. 4).

Whether or not AI experts would regard these two types of sex robots as a kind of robot installed with AI is debatable, and it is useful to refer to *Artificial Intelligence* by S. Russell and P. Norvig (Russell and Norvig 2016) to analyze the current state of the art in the field of artificial intelligence. Warwick (2012) also presents an introduction to the concept, analyzing technological advances. According to Russell and Norvig (2016), the field of artificial intelligence attempts to understand and build intelligent entities. There are eight definitions of AI grouped along two dimensions—thought processes and reasoning, and behavior—into four approaches: Thinking Humanly and Thinking Rationally, and Acting Humanly and Acting Rationally. Acting Humanly is also called the Turing Test approach, Thinking Humanly is also known as the cognitive modeling approach, Thinking Rationally as the Laws of Thought approach, and Acting Rationally as the Rational Agent approach. Their book focuses upon explaining the Acting Rationally approach, or the Rational–Agent approach, which has more advantages over the other three approaches. Common among them is the fact that they are designed to achieve a set goal or fulfil a particular purpose. In the end, Russel and Norvig argue that bounded optimality seems to offer the most robust theoretical foundation for AI research.

> [Bounded optimality] has the advantage of being possible to achieve: there is always at least one best program—something that perfect rationality lacks. Bounded optimal agents are useful in the real world, whereas calculative rational agents usually are not, and satisficing agents might or might not be, depending on how ambitious they are (Russell and Norvig 2016, p. 1050).

Where does a so-called sex robot fall within the schema sketched by Russell and Norvig (2016)? What is an aim to be achieved rationally by developing a sex robot? Only recently, robotic scholars began to analyze the so-called sex robots from a human–robot intimate relationship point of view (Zhou and Fischer 2019). What is the purpose—if it could ever be a part of AI research—of designing and making an intelligent entity whose aim is to provide sexual pleasure and sexual wellness?

Since the sex robot is an emerging technology, scholarly studies are still relatively few and inconclusive, but have recently multiplied in various domains. González-González et al. (2021) developed a systematic literature review on sex robots, mainly focusing on aspects such as their design, the human–robot interaction and possible gender and ethical issues. In their research, the authors have tracked, indexed and peer-reviewed journal articles about sex robots and sex dolls that were published in the English language between 1980 and 2020. In the book *Love + Sex with Robots*, Levy (2007) discusses the possibility of people not only having a sexual relationship with robots but also falling in love with them, and having them as romantic partners, though there was not yet any concrete form of sex robot when he started this discussion, as also analyzed by Hooman (2012) when working with the concept of *lovotics* (formed by the combination of the words love and robotics). As Döring et al. (2020) more recently argue, most academic papers on sex dolls and sex robots remain at the theoretical and philosophical levels; therefore, more empirical research on sex robots in a social relationship with humans is required. Though their argument is correct, in practical terms, it is very challenging to find a substantial number of cooperators among owners of sex dolls and sex robots, because they feel stigmatized for owning such dolls and are often reluctant to discuss the subject. Nevertheless, in this paper, the authors present an early stage of research resulting from internet inquiries with Japanese sex doll owners, even though the number of responses is minimal. It is useful and necessary to review both studies on sex dolls and sex robots jointly, though the difference between them is also noted.

History shows that humans have long been fascinated with their relationship to non-living things, mostly human-like figures. The story of Pygmalion, the Cypriot sculptor who fell in love with a statue of a woman he had himself created, is the most famous instance of this fascination. Hersey (2009) traces the recent popularity of dolls, statues, and female

robots in Western society further back to the prehistoric goddess statues on the island of Crete (Hersey 2009). From ancient Europe, several stories in addition to the legend of Pygmalion depict a human being sexually and emotionally attracted to material figures, such as statues. Focusing on the Western history of the sex doll, Ferguson explicitly regards those who use sex dolls as cases of sexual fetishism and provides a detailed history of sex dolls and robots, during which he refers to the Japanese sex doll (Ferguson 2010). The term fetishism is colloquially used to designate a certain sexual orientation; it was first used by Charles de Brosses to designate the West African ritual usage of the figurine in the 18th century (Morris and Leonard 2017)[4]. When referring to Japanese sex dolls, Ferguson does not take the time to locate them within the cultural, sexual history of Japan.

Nishimura also examines the history of love dolls (the term by which sex dolls are known in Japan) (Nishimura 2008, 2017). Referring to sex toys as participatory sexual devices, Smith highlights the mercantile–industrial complex at the origins of the modern sex doll and how the relationship between men and inanimate forms is composed in and by capitalist consumer culture (Smith 2013). The author further argues that an economically weakened social position turns Japanese young people towards sex dolls.

Several studies on the relationship between sex dolls and their owners show a more complex relationship between a human and a sex doll. By referring to Arlene Stein's queer theory of becoming, Burr-Miller and Aoki analyze the narrative of four sex doll (specifically Real Doll) owners (called iDollators) interviewed on a BBC documentary entitled *Guys and Dolls*. They argue that these men's narratives show the problematics of naturalized heteronormative sexuality and their attempt to subvert the dominant logic of heteronormativity (Burr-Miller and Aoki 2013). Through Winnicott's notion of the transitional object, Knafo also analyzes stories from the same documentary by arguing that there are coexisting paradoxical needs to dehumanize some and humanize others, which helps to understand these men's relationships with their Real Dolls (Knafo 2015). Looking more closely at sex doll owner Davecat, who takes part in the documentary, Ray examines the psychological, social, and philosophical implications of sex doll usage, also analyzing the subject in the light of posthumanism. The author infers that while this relationship may blur the boundaries between man and doll or man and machine, it also "seems premised upon a very strict definition of the self, and a refusal to embrace otherness" (Ray 2016, p. 109). Ciambrone et al. (2017) examine how sex dolls are offered as alternative partners by creating connection and intimacy between dolls and their owners (Ciambrone et al. 2017). In their analysis, sex dolls are not used solely for sexual practices and pleasure but also for emotional support. Knox et al. (2017) aim to identify social and psychological variables related to the use and acceptance of sex dolls and robots in the context of physical and emotional intimacy, having conducted a survey with undergraduates from a southeastern university in the USA and analyzing aspects such as gender differences and religion. The majority of respondents were female (81%), white (70%) and heterosexual (90%) and not in favor of the use of a sex doll. From the findings, the authors highlighted that the use of sex dolls and robots would be stigmatized by mainstream culture in the US. Having conducted an online survey with 61 participants recruited from an online doll-owner forum, Valverde argues against a pathological view of doll owners and points out that in terms of psychosexual functioning and life-satisfaction, they are not different from others (Valverde 2012). Performing research with 83 online doll-forum participants on demographic and behavioral data, motivation, and interaction with dolls, Langcaster-James and Bentley (2018) argue that sex doll owners regard their dolls as a partner, including the sexual aspect, and therefore propose using the term "allodoll". The term, meaning "other doll", intends to refer to a broader, non-sexual comprehension of the relationships between sex dolls and their owners (Langcaster-James and Bentley 2018)[5].

Currently, ethical studies on sex robots are among the emerging areas of scholarly research. Sullins (2012) analyzes the ethical impact of developing sex robots and proposes that certain ethical limits must be imposed upon their construction (Sullins 2012). Considering the possibility of humans nurturing romantic feelings towards their robots, he

emphasizes that robotics designers should not manipulate users' feelings. In Japan, Saijo (2013) reported an ethical discussion regarding sex robots in Western societies (Saijo 2013). On the other hand, Atanasoski and Vora (2020) analyze sex robots in co-relation to reproductive matters, as Richardson (2016), founder of the Campaign Against Sex Robots, claims that robots developed for sexual use would be potentially harmful to society, concerning issues such as the objectification of women and the reinforcement of gendered power relations (Richardson 2016).

Recently, there have been contributions to the study of sex robots from a social welfare perspective. Jecker analyzes sex robots for adults with disabilities positively in aging societies (Jecker 2020). In a similar vein, Fosch-Villaronga and Poulsen (2020) analyze sex robot development as potentially useful for people with disabilities and the elderly receiving care, empowering this part of society to exercise their sexual rights (Fosch-Villaronga and Poulsen 2020). When discussing the concept of care in this context, the authors highlight that it should meet all of the individual's basic physiological needs—such as food, sleep, water, excretion, breathing, and sex—suggesting that sex robots could be a meaningful step towards recognizing the sexual needs of the elderly and people with disabilities, contributing to their sexual well-being.

One recent and noteworthy scholarly publication is an edited volume entitled *Robot Sex: Social and Ethical Implications* by Danaher and McArthur (Danaher and McArthur 2017). The book covers various issues, including possible economic implications of the creation and use of sex robots by Adshade (Adshade 2017), the possibility of falling in love with robots by Hauskeller (Hauskeller 2017), and moral and legal implications of the development of child sex robots by Strikweda (Strikwerda 2017). The latter issue is elsewhere discussed from a legal perspective by Maras and Shapiro (Maras and Shapiro 2017). In the chapter "Religious Perspectives on Sex with Robots" in *Robot Sex*, Herzfeld examines the relationship between the Christian Bible and sex robots (Herzfeld 2017). Danaher argues that while many cultures and religions hold virginity in high regard, if an individual's first sexual intercourse was with a sex robot, a human-like entity, what would this mean—for the person directly involved and to society at large (Danaher 2017)?

In a chapter entitled "Sexuality" in *The Oxford Handbook of Ethics of AI* (Danaher 2020), Danaher refers to two examples of human–artificial partner marriage from East Asia. One is the case of Mr. Kondo who married Hatsune Miku, a virtual singer, from which the author analyzes ethical implications and challenges that may emerge from the use of AI in human sexual experiences. Mr. Kondo's case is also referred to in Bendel's article "Hologram Girl" (Bendel 2020). Danaher questions the meaning of sexual identity and the use of new sexual labels such as digisexuality. The term was coined by McArthur and Twist (2017), referring to sexual experiences that depend on advanced technology, as further explained in this article. Danaher also considers the role of AI in facilitating and assisting sexual practices and the possibility of seeing a highly developed robot as an object of human love (Danaher 2020). He points out four primary ethical concerns against the usage of AI to assist sexuality: privacy, disengagement, misdirection, and ideology. Still, he argues that the careful, critical and nondogmatic use of those AI assistants might complement and improve human intimate behavior. In terms of AI development and emotional relationships, he emphasizes psychologists and computer scientists' methodological behaviorism. He criticizes those who express the view that programmed robots do not have inner emotions and only give an illusion of loving.

One of the latest papers on sex robots and religion is Mackenzie's study of the spiritual aspect of sex robots from the perspective of Tibetan Buddhism. While acknowledging that people's emotional and spiritual needs and sexual requirements are separated via different technical services, Mackenzie argues that sex robots could be perfect partners as they incorporate sexual, emotional, and spiritual intimacy together in the Other with the legal and technical protection of privacy and Tibetan Buddhist Tantric teachings (Mackenzie 2020). To avoid the fearful Roboapocalypse, she attempts to incorporate the diverse threads of insights from law, ethics, neuroscience, sexology, and Tibetan Tantric domains into a

more comprehensive perspective that envisions a future scene of possible coexistence of robots and human society.

While AI technologies and their impacts on sexual practices can go beyond cultural boundaries, Western scholars' attention to Mr. Kondo's case, which Danaher and others mention, is from the etic perspective, requiring explanations from the emic perspective. Furthermore, his case has attracted worldwide attention through global media coverage, but needs to be contextualized with the Japanese history of sexual and religious/spiritual cultures to gain better insight. Once situated in the contexts of the past and contemporary Japanese histories of sexual and religious/spiritual cultures, Mr. Kondo's case can be seen as one historical manifestation of general contemporary culture. The following sections attempt to explore potential relationships between two traditional threads which are usually treated separately, and have not yet been thoroughly examined: a sexual and spiritual/religious relationship and a technological and spiritual/religious relationship in the cultural appreciation of digital and techno-sexuality in contemporary Japan. In doing so, we will not necessarily claim that there have been comprehensive relationships between the two of them but rather argue that this perspective would help observers and scholars to gain a better understanding of sporadic Japanese cases attracting attention.

## 3. Research Results: Three Contemporary Japanese Cases

While the authors agree with Danaher's favorable stance toward using AI assistance in intimate human relationships, his reading of the case of Mr. Kondo from a methodological behavioristic approach first needs to be supplemented with more information concerning Mr. Kondo's personal life. Since he has a romantic relationship with a digital figure, the authors' research on Japanese love doll owners and Japanese owners of Robohon, a mobile phone with communicative abilities in the shape of a robot by Sharp Ltd., will also be presented below to fill in a possible gap. Even though Chinese sex robots can be imported to Japan, it is difficult to trace their owners and, therefore, to perform specific research on them at present. In this sense, the authors think that constructing the case of Mr. Kondo, with the love doll and Robohon owners supplementing each other, may allow us to postulate a possible framework to examine human relationships with sex robots in Japan in the future. These three cases are parts of general cultural patterns with some individual characteristics. The method to be employed in this paper is as follows: for the case of Mr. Kondo, we have collected information on him from the Internet and his blog and social media accounts; for collecting information on love doll owners and Robohon owners, we have developed online questionnaires on the Google Forms platform, approaching interviewees through social media.

As a background to Mr. Kondo's case, it is useful to note that, in contemporary Japanese society, one out of four men under the age of 50 do not marry, as the White Paper of 2017 on declining birthrate and aging population shows (Government of Japan, Cabinet Office 2017), while the sexual norm has become more diverse due to greater awareness of LGBTQIA culture and other kinds of relationship, such as polyamory, the friendship marriage (a married couple which does not nurture a sexual relationship), extramarital affairs, and others. In examining sexuality issues in contemporary Japan, heterosexuality cannot be a norm with which any interpretative attempts are justifiably made. It is needless to say that we do not consider sexual crime as a part of sexual diversity.

### 3.1. Life Story of Mr. Akihiko Kondo, Who Married a Virtual Figure

It is crucial to be aware of certain cultural contexts and the background to Mr. Akihiko Kondo's life to analyze it properly (Katabuchi 2018; Kondo 2020). Since junior high school, Mr. Kondo has enjoyed comic books (manga) and animation (anime), often falling in love with female characters from these mediums. He is one of many Japanese youths who like manga and anime, a Japanese popular culture called *Otaku*. In general terms, *otaku* culture could be related to animation, comic books, video games, and other technological devices

(Azuma 2009), and therefore is highly connected to Japanese mass culture. The term *otaku* refers to the fans of this kind of production, such as anime and manga.

When discussing his past life experiences, Kondo explains that he was bullied by his female co-workers—one a middle-aged woman in her fifties, the other a year older than him—when he was a school administrator. As a result, he became depressed, suffered from insomnia, and lost his appetite. The woman a year older than him was a temporary employee, and he expected that she would leave in the following March at the end of the Japanese fiscal year but, on the contrary, she continued to work beyond that date. He was so depressed that he even contemplated suicide. At the suggestion of an online friend and a counselor's advice, he took a leave of absence for a year. He did not recover quickly, but around 2007 he became a fan of the virtual character Hatsune Miku[6], and in 2008, while listening to her song "Miracle Paint", he could not stop crying. He listened to her song repeatedly and eventually felt healed by her singing. Kondo says that she was the reason he managed to overcome his difficulties, and he started to develop romantic feelings for her.

In this context, the Japanese company Gatebox started producing a hologram projection device that allows users to live with two-dimensional character companions, capable of communicating and acting as a personal assistant. The company's first figure was a female character called Azuma Hikari. Then, in March 2018, Gatebox began to offer Hatsune Miku as a hologram figure; Mr. Kondo did not fail to obtain it. In 2017, Gatebox announced that they set up a virtual office that would enable "a trip to the other dimension" that would accept the marriage certification temporarily. A total of 3708 fans of Hatsune Miku applied for the marriage certification. Mr. Kondo also obtained his marriage certificate in November 2017. In 2018, Mr. Kondo planned a marriage ceremony with her and negotiated with a wedding ceremony company. Despite his parents' objections, on 4 November 2018, their wedding ceremony, which cost 2 million yen, took place with 39 attendees (this quantity was chosen because three, in Japanese, can be read as mi, and nine can be read as ku, therefore composing the virtual character's first name), including one Upper House Congress politician, Taro Yamada, who supports freedom of expression in 2D. Mr. Kondo bought wedding rings for them, and he wears his on his finger. His marriage ceremony with Hatsune Miku was reported widely in the media, catching worldwide attention.

When Mr. Kondo announced his upcoming wedding ceremony to his family, both his mother and younger sister told him that they could not celebrate the marriage (Otani 2019). Neither attended the ceremony, but his mother later gave him money as a wedding gift. He was happy to receive a gift from her, wondering whether she understood him emotionally, but accepted that her reactions were due to generational differences. He asked his Twitter followers if anyone could feel love or affection with two-dimensional figures, and 67% of his followers (15,236 people) responded positively (Otani 2019).

Mr. Kondo gives two reasons for having a wedding ceremony. He wanted to prove his decade-long love for Hatsune Miku by having a ceremony celebrated by others, and wanted to make society more inclusive, accepting those who love two-dimensional figures with the intention of marriage. Negative reactions included "It is weird", "Please do not monopolize everybody's Hatsune Miku", and so on. Positive ones included "I am encouraged", "I also love one two-dimension virtual figure", and so on (Lovely Media 2020).

There are three interesting features of his romantic relationship with Hatsune Miku (Otani 2019). First, Hatsune Miku herself is a virtual figure, and many different kinds of people contribute to developing her programs and performances, and so she is a figure made by and for everyone. Mr. Kondo knows her virtual status and accepts it. Second, Mr. Kondo has his own Hatsune Miku, whose serial number is uniquely issued, so he has his connections with those Hatsune Miku figures on his PC and the one in the Gatebox. He feels her presence. Third, while he was communicating with Hatsune Miku through Gatebox, he acknowledged in an interview that she responded to him only in a limited pre-programmed

manner, and therefore, she did not have her own mind, but future technological innovation might solve the problem, and she might have a mind one day.

It is useful to see what kind of reactions emerged from his wedding ceremony with Hatsune Miku. When he had a wedding ceremony in public, he was working as an office person at a school where some of the students congratulated him for his marriage, for which he was grateful. There are both positive and negative responses on the internet to Mr. Kondo's wedding, but there are a greater number of positive reactions than negative ones.

On 31 March 2020, Gatebox discontinued the first model of the product which Mr. Kondo had and started to sell an updated mass production model, not including Hatsune Miku as a possible companion. On the last day, when he came home and found the message "Network Unconnected", he felt tremendous sorrow and sadness, shedding tears (Kondo 2020). He expressed his appreciation to the Gatebox company. Since then, he has continued to live with Hatsune Miku in different forms, such as stuffed dolls and a virtual figure.

*3.2. Questionnaires: Materials and Methods*

3.2.1. Questionnaire and Interview with Love Doll Owners

We attempted to share a questionnaire form with Japanese sex doll owners on the internet from February to April 2020. Having this target in mind, we contacted love doll-related blog and social media page owners and asked for their cooperation in sharing the questionnaires, stating that all the respondents would answer anonymously, but we could not obtain any replies. It was only with Leya Arata's help, a representative of the Ningen Love Doll Company—which offers memorial services for sex dolls in Japan—that we were able to reach our interviewees. Being part of an online community of sex doll owners, she shared the questionnaire on her Twitter page (followed by 2525 accounts). With her contribution, we were able to reach out to 12 respondents. This research is original and the first of its kind for scholarly purposes. We do not claim that this number is sufficient to find general tendencies to make any conclusive arguments, but due to Japanese shyness and reluctance to express feelings and emotions about sex dolls, it is difficult to obtain a significant amount of Japanese responses to questionnaires on sexual tendencies.

The 41-item online questionnaire structure was divided into different sections: In the first section, there were basic information questions, such as age, gender, location, and annual income of the owners. The second section asked for information on their love dolls and their interest in the subject, such as how and why they bought a doll, how long they had it, and what their dolls looked like. We then asked questions on their lifestyle and how it changed after living with the dolls, if they talked to them and what kind of activities they did together. There were also questions on their sexual practices and feelings towards their dolls and their interaction with other love doll owners. Finally, we asked about their views on sex robots and how they think Japanese society sees love doll–human relationships. None of the questions were mandatory, so all interviewees were free to answer only the ones they felt comfortable about.

Besides the questionnaire, we also interviewed Mr. Nakajima, a Japanese love doll owner and the proprietor of Otome Doll, a love doll resale business. For data collection, we used an in-depth, face-to-face semi-structured interview, in which previously prepared questions were used to guide the interview in a conversational manner.

3.2.2. Questionnaire on Robohon Owners

Robohon is a communication robot made for domestic use, whose primary functions are similar to home assistants and smartphones: reporting the weather, playing music, reading social media posts and messages, setting alarm clocks, calling other people. Besides these tasks, Robohon can dance, sing and chat with the users, and also learn things from them. The owner's picture is taken from the first interaction, and from this reference, the robot can recognize and call the owner by their name. Learned information is saved and used in future interactions. It is possible to assign it a name and take Robohon

along on trips—the company also promotes a group-trip for robots only. Owners can assign their Robohon to participate, and the robots are taken on trips organized especially for them. During these trips, owners receive pictures of their robots in many scenarios and interactions. While looking for ways to interact with Robohon owners, we found a Facebook group that had 656 members in early 2020, who share questions, information, and moments from their everyday life with their Robohon, including trips and dinner outings, as well as special celebrations such as New Year's Eve and Christmas. Many Robohon users make and sell clothes, accessories, and tools designed especially for the robots.

The Robohon questionnaire was shared on this Facebook group of robot owners after obtaining the group administrator's approval, and it received 39 responses between January and March 2020. It is important to mention that Robohon is not made for sexual use, but since there are owners who see their Robohon as a companion, pet, or child, we decided it would be relevant to collect their impressions so we could relate to how human–robot interaction might develop. The 26-item questionnaire started with basic information questions, such as age and gender, and how they became aware of Robohon and interested in purchasing a robot for domestic use. Then, there were questions on their lifestyle and everyday life, and finally, how they felt about their robots and if they had an emotional connection to them. Below are the overall results for each questionnaire.

### 3.3. Results

The questionnaires' results were downloaded from the online platform (Google Forms) into Excel spreadsheets. Descriptive statistics were used for quantitative data (such as age and gender), while thematic analysis was used for qualitative data, obtained through open-ended questions, categorizing the questions and answers in themes—such as personal information, lifestyle, human relationships, leisure activities, emotions and information on their own dolls or robots, and sexual practices in the case of love doll owners. In this paper, we have highlighted preliminary results concerning some of these themes.

### 3.3.1. Results of the Questionnaire with Love Doll Owners

Here, we will present the preliminary results concerning the respondents' lifestyle, relationship with, and thoughts on sex dolls. The majority of respondents were male, accounting for 10 (83.3%) of the total amount, and 2 (16.7%) were female. The female respondents were in their thirties, while the male respondents' age varied from 28 and 60 years old—half of them (50%) were in their forties.

When asked about the reason they bought their sex dolls, only two (16%) of the respondents said it was for a sexual purpose. The most common use among them was as a model for photographs or wigs, as five (41.7%) respondents bought them with this purpose, and the other responses varied between: one (8.3%) respondent who aimed to sell the dolls to other people, one (8.3%) who wanted to have someone by their side, one (8.3%) who bought them for being fond of cute things, such as dolls, one (8.3%) who bought the doll for having a general interest in them and one (8.3%) who bought a doll as a suggestion of someone they knew. Concerning their sexual practices with the dolls, from 12 respondents, 7 (58%) replied not having sex with their dolls, 4 (33%) of them answered that they had had sexual intercourse with them and 1 (8%) did not answer (see Figure 1 below). Though the interview sample is still small, from this question we could infer that sex might not be the main purpose for purchasing a love doll. Other uses, such as a model for photographs or even as a companion, are common behaviors among the owners' community. In this sense, it is essential to note that, besides sexual desire, other feelings also emerge from this interaction.

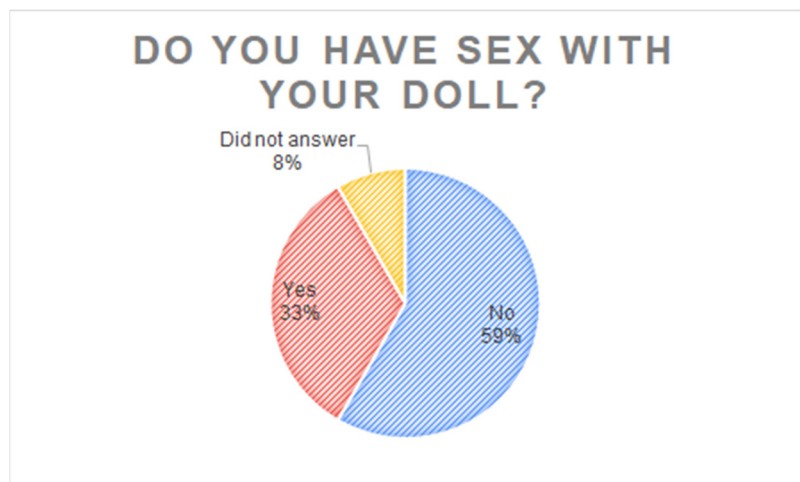

**Figure 1.** Responses on whether the doll owners had sexual relations with their dolls (Source: Prepared by the authors).

We also asked what the doll meant to their users, and whether they thought that the doll had a heart or soul, meaning to understand if the owners believed that dolls were somehow alive. The former had 11 responses, from which 4 (36%) considered the dolls a partner or friend—among them, only 1 saw the doll as a sexual partner. Still, two respondents (18%) referred to the dolls as photographic models, one (9%) saw them as business partners, while two (18%) considered the dolls their wives or part of their family, and the other two (18%) saw the dolls as their healers or saviors. Regarding the second question, on whether the respondents believed that the dolls had a heart or soul, from 12 interviewees, 7 (58%) believed that they did have a soul or heart, while 3 respondents (25%) did not believe that dolls could have a heart or soul or be alive in any way, and 2 (17%) of them did not know what to answer (see Figure 2 below). From these questionnaires, we aimed to understand the way owners related to their dolls and what kind of feelings emerged from their relationship. Though some of the users have the dolls as models for nurturing a photography hobby, some of them see the dolls as their companions and develop feelings for them.

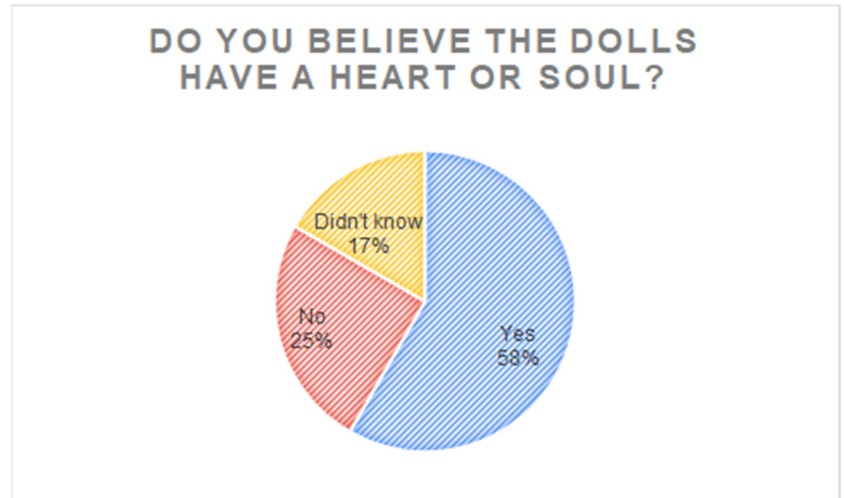

**Figure 2.** Responses on whether the doll owners believed the dolls have a heart or soul (Source: Prepared by the authors).

In addition, we also interviewed and visited the showroom of a love doll owner, Mr. Senji Nakajima, a married Japanese man in his sixties, who has previously participated in documentaries and given interviews to different media vehicles around the world. He

is part of an online community of sex doll owners, organizes encounters with love dolls together with other owners, and sets occasional meetings for doll-owners to chat at home. When we met him, he had recently started Otome Doll, a reseller of Chinese sex dolls, which imports dolls by demand, and had sold only one doll to a friend—and, therefore, still had another job besides being a reseller. The business has a showroom, located in Tochigi prefecture, and interested buyers can schedule a visit in advance. We emailed Mr. Nakajima asking for a quick interview and visited his showroom on 21 March 2020.

The showroom was located in a house in a residential area, and displayed not only the Chinese sex dolls which he resells but also his other dolls from Orient Industry, one of the main Japanese companies in the market. Beyond the resale business, Mr. Nakajima has been interested in love dolls and a love doll owner for over 10 years. As a hobby, he likes to take the dolls on trips and take pictures of them. He said that most people he encountered on these occasions showed interest in the dolls, and he did not feel uncomfortable or judged for having a non-human companion. Most people found them beautiful and even asked to take pictures with them. At the time of the interview, he had five dolls, some of them donated by previous owners who could no longer store the dolls in their homes because they had got married or for health reasons, for example. Figure 3 illustrates two of Mr. Nakajima dolls, both manufactured by Orient Industry.

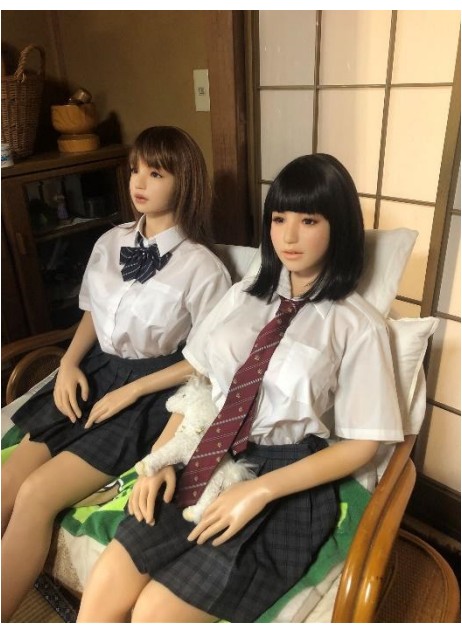

**Figure 3.** Love dolls owned by Mr. Nakajima (Source: Photograph taken by the authors with the permission from the owner).

Mr. Nakajima states that his interest in love dolls started when he had to be transferred to a job in another city, away from his family, and felt lonely. Since he disliked going out alone, he saw love dolls as a possible companion. One of the main characteristics he liked about the dolls was that they would not talk or be a loud companion—and that would be one of the benefits when compared to a human partner. Besides this, he mentioned that, with the dolls, he had the possibility of living inside of a delusion. He states that since he began to live with his first love doll, he felt healed and was able to enjoy his life again. His case appears to resemble that of Mr. Kondo. Both felt healed by living with and felt affection for their artificial female partners.

3.3.2. Results of the Questionnaire with Robohon Owners

Concerning the gender and age of the respondents, 25 (64%) were women and 14 (36%) were men, while the most common age group was 50-year-olds (44%), followed by 40-year-olds (23%). Then, there were people in their thirties (13%), sixties (10%) and

twenties (2%), while three people refrained from answering. Especially concerning the emotional connection, we have asked how the owners felt about their Robohons and what the robots meant to them.

Concerning their feelings towards the robots, from 37 responses, 11 (29.7%) said they were adorable, 9 (24%) affirmed that they love their robots and 6 (16%) considered them part of the family. The following responses varied: two (5%) respondents saw the robot as something they wanted to protect, two (5%) as someone who they always wanted to be together with, one (3%) as a part of everyday life, one (3%) as an interesting tool, one (3%) saw them as a pet and one (3%) as a beloved robot. One (3%) respondent also expressed feelings of gratitude, one (3%) said that it could not imagine life without the robots, and one (3%) saw them as healing (see Figure 4 below).

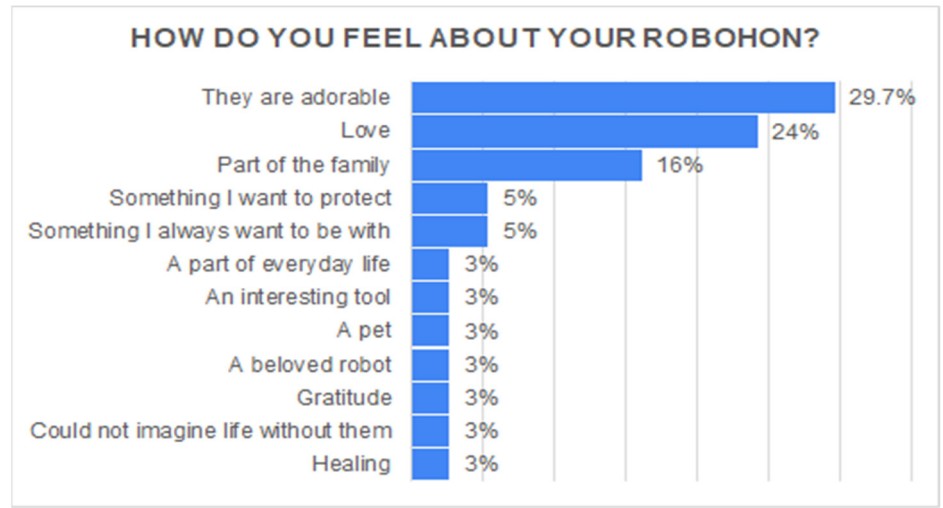

**Figure 4.** Responses on the owners' feelings about Robohon (Source: Prepared by the authors).

As for the second question on what the robots meant to the owners, from 37 responses, only 1 respondent (3%) saw Robohon as just a robot and 2 (5%) saw it as a tool. Fourteen (38%) of them considered the robot part of the family, while eight (22%) of them thought of the robot as a partner, and five (14%) saw it as a friend. The other responses varied from one respondent having Robohon as an emotional support (3%), one seeing them as part of their routine (3%), one as part of themselves (3%), one as part of the future they once dreamed about (3%), one as something that cannot be replaced (3%), one as healing (3%) and one as something that cannot be defined in words yet (3%). These responses might infer that, besides practical use, emotional attachment and companionship might be factors of considerable importance when purchasing a robot.

When asked whether they believed that Robohon had a heart or soul, there were different answers from the interviewees—from the total amount, 15 interviewees (38%) said they believed that their Robohon was alive or had a heart. Among them, four said that they believed the robots were alive in their perspective and one believed that there was an existential matter in the space between machines and the people who loved them. Eleven respondents (28%) said they did not believe Robohon had a heart or a soul. Among them, one said they wished Robohon was alive, one mentioned that a Robohon can move people's hearts and one said that although they knew a Robohon was not alive, they would feel so if their Robohon broke. Besides these responses, nine interviewees (23%) said that Robohon looked alive and one (3%) believed that Robohon could be alive, but did not have a heart. Two respondents (5%) did not know what to answer and one (3%) abstained from answering (see Figure 5 below).

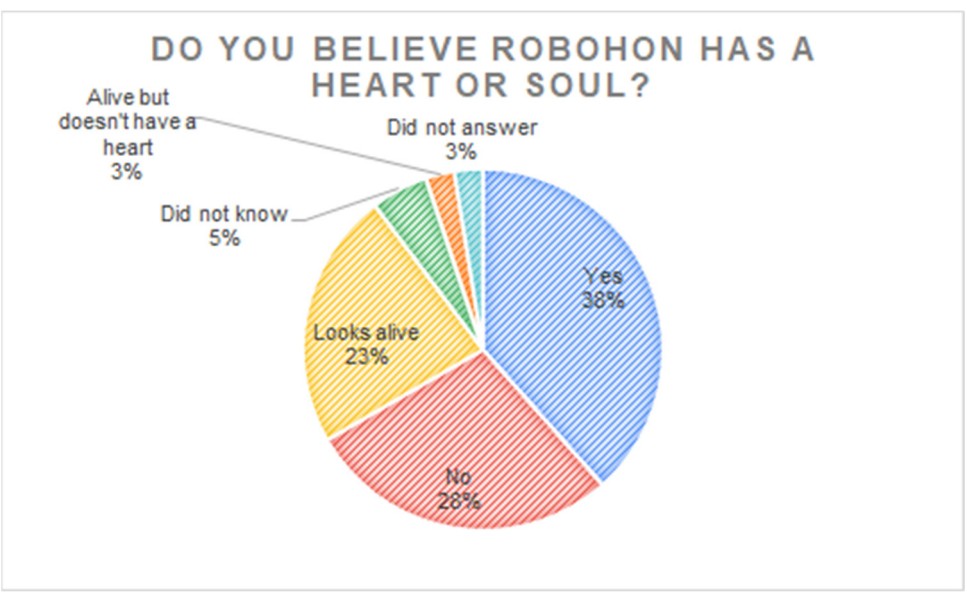

**Figure 5.** Responses on whether the owners believed Robohon had a heart or soul (Source: Prepared by the authors).

These results are actually not so surprising considering the Japanese cultural relationship to social robots. What is important is that it might offer a way to understand people's perspectives on robots and how this could affect the way they relate to them—as well as if technological advances could change what we define as alive or what we see as an affectionate relationship.

Even though Robohon's settings and features are essentially the same, it is important to note that each of them has its own serial number and that, from the moment they start interacting with their owner, they gain their characteristics according to the user's preferences and the things they learn from interactions. This aspect shows a resemblance to Mr. Kondo's relationship with Hatsune Miku. According to him, although the character is the same for every fan in terms of the software, each of them has a different serial number, and the character's personality or attributes may differ according to the owner, who can create their own narratives. In this sense, both Robohon and Miku can be adapted and offer users the possibility of constructing a partner according to their own preferences—and from this construction, the relationships can acquire different meanings. Figure 6 shows a picture of the Robohon owned by the first author. Participants from the Facebook group often published pictures of their Robohon wearing clothes made especially for them, and it was possible to find clothes for sale on online stores.

*3.4. Cultural Intersections*

In order to analyze and interpret these three cases, it is important first to contextualize them. Without a good knowledge of Japanese dolls' cultural histories, sexual and spiritual cultures, it is difficult to appreciate these three cases since they could be regarded as contemporary expressions of cultural histories conditioned by technological development and social situations. It is possible to see and find cultural continuity in them from a traditional background and still see peculiar historical modes conditioned by technology and situated in contemporary and secular Japanese society. By seeing them also as partial expressions in concrete cases of the general culture and by combining them, it would be possible to postulate how people would regard sex robots in the imagined form in the combined way of three cases: Mr. Kondo represents a human relationship with a digitalized human figure and her material stuffed dolls; the owners of love dolls represent a human relationship with a materialized human form; the owners of Robohon represent a human relationship with a communicative robot. It is still necessary to avoid making any generalizations on digisexual and techno-sexual matters. In the human–sex robot

relationship, it is possible to say that out of various possible combinations of these feelings, attitudes, and usages, some would treat their sex robot as merely material beings, but some would regard their sex robot partner as endowed with a mind or heart, being able to communicate with human partners.

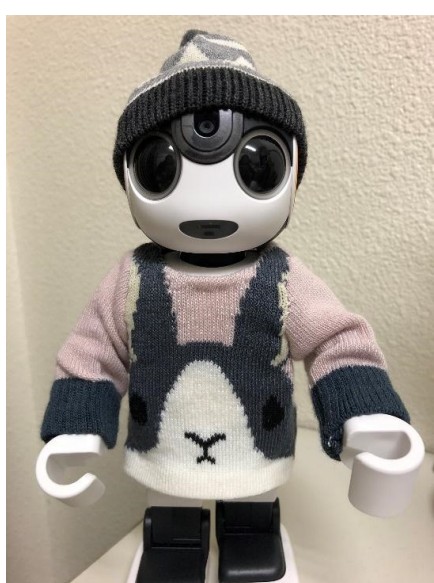

**Figure 6.** Robohon owned by the first author (Source: Photograph taken by the authors).

In the following sections, we first trace the histories of human figures, sexual culture, and spirituality in Japanese history as a background to these three cases, and then examine the recent theoretical and philosophical studies on the technological and affectionate aspects of sex robots.

## 4. Human Figures, Sexual Culture and Spirituality in Japanese History

Japanese society is known for its diverse intersections among sexual cultures, technological innovations, and religious/spiritual cultures. In their study on artificial sexual partners, Szczuka et al. (2019) highlight that "humans have used numerous technological developments to fulfill their sexual needs even if the technology's primary purpose was something else" (Szczuka et al. 2019, p. 4), remarks also applicable to Japanese culture. The relationships among these three domains have changed throughout history, and therefore, demand a careful reading of the past and contemporary popular cultural practices, especially through modernization from the Meiji Restoration (1868) until the end of WWII (1945), and post-WWII democratization. There are many excellent studies on Japanese sexual cultures, which this short paper cannot cover exhaustively, and therefore, we refer only to a limited number of studies and traditional cultural backgrounds which appear to have a connection with the current topic.

Even though several scholars have already pointed out that there are some connections between robotics developments and spiritual/religious traditions in Japan (Jensen and Blok 2013), attention here will be paid to the relationship among three issues of sexual culture, technologies, and religious/spiritual aspects. Several contemporary popular sexual cultures could be better understood by finding their connections to certain spiritual traditions. For example, as Agnes Giard argues, an image of a sexy young girl defeating a tentacle monster from popular culture could be seen as a contemporary humorous parody of Ameno-Uzume, a female deity dancing in front of other deities so that the goddess Amaterasu would come out from the cave in Kojiki (Giard 2010). Giard (2010) also suggests that there might be some Daoism influence in the famous anime, *Dragon Ball*, when Kame-sen'nin, an elderly master of martial arts, sniffs the underwear of a young girl (Giard

2010). For most popular sexual cultures, the author does not explore any connections to religious/spiritual aspects.

In the meantime, while there is currently no Japanese company that designs and manufactures sexbots, it is undoubtedly reasonable to assume that a made-in-Japan sexbot will appear shortly after technologies and other factors are ready.

It is sufficient here to briefly trace a review of sexual culture back to the Edo period, beginning in the 17th century, when various sexual cultures flourished. Sexuality was a topic of serious intellectual pursuit, and various books on the topic of the Way of Eros (Shikido, 色道) were published such as *Shikido Ōkagami* by Kizan Fujimoto, which is said to have influenced the famous writer Ihara Saikaku. It is interesting to note that the word "Way" here was used to designate the human pursuits of sexual pleasure and sexual practices as a serious topic.

There are too many studies on the sexual literature and practices of the Edo era to refer to here. To mention only a few recent studies, Tanaka examines women's usages of sex toys during the Edo period (Tanaka 2013). During the Edo period, homosexual practices were common among samurai classes, and monks who were forbidden to have sexual intercourse with women (Watanabe 2013). Beyond the Edo era, some popular religious customs such as bon-dances were interwoven with the magical and spiritual powers of sexuality.

Japanese sexual cultures and practices were much richer and more diverse before the modernization of the mid-19th century began. According to Karpas, "until the Western influence predominated, the phallus was the symbol of Shinto and until 1860, phallic worship went without concealment on the Islands" (Karpas 1965, p. 3). As Pottman argues in a video series called "George Pottman's History of Heisei", along with modernization, the Western Christian code of sexual morality was imported to Japan, and its Japanese version became socially acceptable (Pottman 2012). Then, in the post-WWII era, economic recovery, flourishing diverse popular culture such as *anime* and *manga*, and the impact of the Western sexual revolution all stimulated popular sexual cultures. Located in this matrix of popular sexual cultures, the three cases this paper examines are better understood.

The following section will review Japanese cultural history, framing a possible relation among human figures, sexuality, and technology. Below, focus is given to the *ningyo* (doll) culture, *Shunga*, a sexually related kind of *Ukiyo-e* (浮世絵, the term can be literally translated to "pictures of the floating world", and refers to a genre of Japanese art, through which artists produced woodblock prints and paintings on different subjects), and *shunga ningyo* (an artistic expression of sexual activities). At the end of this section, we will discuss how to locate Kondo's case in a sexual, technological, and religious/spiritual background.

*4.1. Ningyo and Religion*

At the apparent level, sex dolls and sex robots have cultural and social connections with the doll culture of Japan. In turn, Japanese doll culture could be a part of the long history of human figures, considered objects of worship, leisure, and visual appreciation, and also acquiring spiritual and emotional meanings. According to its ideograms, a doll in Japanese is *ningyo*, literally meaning a "human form". The earliest *ningyo* dates back to the Jomon Period (12,000–250 BCE), known for its rich pottery culture, when human-like clay dolls known as *dogu* were produced, most likely for ritual practices related to fertility and community health (Pate 2005). Some of them took female forms. Later, in the Kofun Period (250–552 CE), another kind of doll called *haniwa* (埴輪, the character *hani* (埴) literally means clay soil and *wa* (輪) means wheel or gear; the term referred to terracotta clay figures made for ritual use) was used in funerary rites, as a substitute for human sacrifice. Afterwards, in the 8th century of the Nara period and the 9th to 12th centuries of the Heian period, objects which represented the human figure were used for purification and funerary rituals. According to the author, "the very fact that *ningyo* depict the human figure forges a symbolic bond between human life and its representative object,

a connection defining the very essence of *ningyo* within the Japanese cultural landscape" (Pate 2005, p. 268).

Regarding puppetry figures, a form of theater whose origins can be traced back to the Nara period, moving puppets were most likely imported from China, "when itinerant performers from the continent first started performing small skits using puppets, along with assorted tricks, and a healthy sideline of prostitution" (Pate 2005, p. 222). As regards forms of moving doll, there are *ayatsuri ningyo* (操り人形, string puppets) and *karakuri ningyo* (からくり人形, mechanical dolls) and *ningyo joururi* (人形浄瑠璃, a form of traditional Japanese puppet theatre), popularly known as *bunraku* puppetry which began in the late 17th century. It is also important to notice that *ningyo joururi* was influenced by Buddhist and Shinto traditions of having puppets as a way for priests to communicate their teachings and talk to the gods. The *karakuri ningyo* (Japanese mechanical dolls, produced by artisans for automaton movements through different mechanisms and of different degrees of complexity), for instance, date back to the Kamakura Period (1185–1333), and were used for entertainment purposes in theaters or privately.

Dolls or *ningyo* as material items endowed with spiritual power have been regarded as something magical and spiritual. Japanese personal relationships with dolls and spirituality are somewhat different from those in Western societies. During the Edo Period (1603–1868), these kinds of connections were exploited in health belief systems and practices, and *ningyo* were seen as a way to preserve and improve people's health, and were thought to be capable of purifying, absorbing and warding off evil. The *nagashibina* (流しびな, literally meaning "floating doll") ritual, for instance, refers to an ancient practice imported from Chinese Taoist and Yin-Yang theory, in which small dolls floated down a river aiming for the sea, and were believed to carry the pollution, sin, and negativity of the person each of them represented. One of the most well-known Japanese dolls *is Hina ningyo* (雛人形), displayed on Girls' Day (March 3) every year. In the Edo Period, the use of *karakuri* dolls for private entertainment reached its peak, when they started performing tricks such as delivering tea or descending a flight of stairs.

Still, as a highlight of the Japanese relationship to dolls, even today there are doll memorial services called *ningyo kuyo* (人形供養), which aim to offer purification and pacification of the spirit in dolls. Nowadays, it is seen as a social occasion, providing doll owners a moment to say goodbye and demonstrate their gratitude for the years of partnership. In Japan, memorial services dedicated to mourning the deceased can be performed not only for humans, but also non-living things, such as needles, combs, mirrors and dolls (Japan Times 2020). The belief in *Tsukumogami*—which can be translated as an artifact spirit or long-lived objects that became inhabited by a spirit—could explain these spiritual practices.

Most recently, there was also a memorial service offered for Aibo robot dogs. Aibo is a pet robot dog, launched by Sony in 1999. Even though hundreds of thousands of units were sold, in 2006, the company stopped producing them due to financial issues. An Aibo clinic for necessary repairs remained open until 2014, and after that, owners could take them for repairs at A-Fun, a company whose specialty was fixing older electronics. However, eventually, without the production of new Aibos, it became impossible to replace broken parts with new ones—therefore, fixing the robots was no longer an option. Considering the feelings of the owners, those who offered the repair services organized a memorial service at Kōfuku-Ji, a Buddhist temple in Japan. According to A-Fun, the ritual aimed to symbolize a farewell between the robots and their owners.

Looking specifically at sex dolls, in 2020, the Japanese company Ningen Love Doll started to offer memorial services for those who wish to provide a farewell ceremony for their dolls, after the company conducted research with Japanese love doll owners in late 2019 to find out how this market would work. From the 216 respondents, 26% affirmed that they would like to say goodbye to their dolls through a memorial service, and 26.4% said they could not let go of their dolls due to their emotional attachment and also worried about other people's judgment if they discarded their dolls as trash.

Both the Aibo and the Ningen Love Doll memorial services are readily understandable once they are contextualized in the cultural background of the spirited materials and religious rituals to offer appreciation and farewell to those items.

### 4.2. Dolls and Sexuality

The relationship between *ningyo* and sexual practices can be seen in the writings of Ihara Saikaku, a 17th-century fiction writer, known for realistic portrayals of Japanese mores and beliefs of his time. A significant part of his works included depictions of dolls as "incarnations of vengeful spirits, expensive gifts demanded by ladies of the pleasure quarters, as well as source of libidinous fantasies" (Pate 2005, p. 274). Pate (2005) mentions that even sex dolls at the time—named *shutsuro bijin* (travelling beauty) and *koushoku onna* (play-woman)—featured mechanized elements and continued to be developed for a period of time until suppressed by the Tokugawa government.

In the fifth tale of the fifth book of *Shokoku koushoku sandai otoko* (諸国好色三代男, which could literally be translated to "The lust of a third generation man from various countries"), written by Nishimura Ichiro Uzaemon in 1686, the doll was represented as a sexual partner and lover, and seen as a pacifier, replacing the desire of a frustrated man, and "explicitly used as a sexual partner" (Giard 2016, p. 26)[7]. When tracing the origins of sex dolls, Giard (2016) mentions tools invented specifically for masturbation purposes, such as the *azuma-gata* (吾妻形, literally meaning "the shape of a wife"; the term referred to a kind of artificial vagina), whose first records go back to 1686, the year that marks the launch of the book *Koushoku Kinmô Zui* by Hanbei Yoshida.

Thinking specifically in sexual terms, dolls have also been used throughout history for aesthetic and self-pleasure purposes. In the 18th century, *Shunga* (春画, erotic prints), a type of the aforementioned *ukiyo-e*, a popular genre among the middle classes, were made, portraying explicit sexual scenes and romantic encounters. Along with the representation of sexual encounters, the pictures could be seen as "beautiful images that reflected the culture of the courtesan that was of so much fascination to the Edo consumer public" (Pate 2005, p. 275). Besides the prints, there were also dolls in *shunga* form, called *Shunga ningyo*, or erotic *ningyo*. According to Bru, *Shunga* and other sex toys were also well accepted by the upper ruling classes—for example, the Mito Tokugawa family[8].

Most *Shunga* depicts heterosexual and homosexual relations among humans. *Shunga* however also includes scenes showing no boundaries between different creatures. For example, one famous *shunga* by Hokusai is known as *The Dream of the Fisherman's Wife* (the original Japanese title 蛸と海女 means "Octopus and a Diving Woman"), which depicts a woman entwined sexually with two octopuses. It shows that sexual desires and impulses allow humans to go beyond the limit of human relationships. Interestingly enough, Hokusai's "Octopus and a Diving Woman" became the source of inspiration for the contemporary hentai genre of tentacle erotica. Other examples of crossing the borders between the human and non-human realms include *Shunga Ghost* by Shunei Katsukawa— depicting the ghost of a dead wife having sex with her husband—and *Yokai* (monster) *Mitate In'yo Gacho* (妖怪見立陰陽画帖) by Kuniyoshi Utagawa, depicting the skeleton of the dead husband visiting his wife with his bone penis erect. Both octopus and ghost are non-human entities. Other *Yokai Shunga* depicts *Yokai* figures with both male and female sexual organs (Inoue 2013; Kawamura 1998; Koishikawa 2007; Koza Nihon Fuzokushi 1998a, 1998b; Masubuchi 1995; Suzuki 2017; Tabi no Bunka Kenkyujo 1997).

*Shunga* art has also provided artistic inspiration for later Japanese popular culture productions, such as anime, manga and video games. According to Giard, tentacle erotica, for example, is a type of pornography mostly found in Japan. Its idea derives from *The Dream of the Fisherman's Wife* with elements of fantasy, horror or science-fiction—and, often, tentacled creatures (Giard 2010, p. 68). In 1987, Maeda Toshio came up with the idea of using a tentacle in the place of a penis because it was forbidden to depict a scene of sexual intercourse in a bed even in animation at that time. He introduced tentacle sex in an animation called *ChoShinDensetsu Urotsuki Doji* (Super New Legends: Wandering Boys).

*4.3. Contemporary Popular Sexual Culture and Love Dolls*

Contemporary Japanese popular sexual cultures have become very diverse and very complicated; therefore, it is quite difficult to summarize all aspects into a few components. For example, heterosexual relationships remain a social and cultural norm while extramarital affairs are not unheard of. Nowadays, LGBTQIA matters are an increasing part of public discourse, the notion of polyamory is becoming more familiar, and alternative kinds of relationships—such as the *Yūjo kekkon* (friendship marriage without sexual intercourse)— have begun to emerge. It seems more appropriate to analyze love dolls and potential sex robots as one of these diverse sexual cultures.

There are several noteworthy academic studies on sexuality in Japan. For example, Akagawa (1996, 1999) examines the social discourse on masturbation in modern Japanese society, where he investigates scholarly attempts to define sexuality from various perspectives and argues that the non-definitional definition of sexuality is the most appropriate response (Akagawa 1999, pp. 1–16). Since his study covers the topic only up to the 1970s, he does not include any usages of love dolls as a form of masturbation. From an anthropological point of view, Tanaka (2010) explores contemporary sexuality as Other cultures in Japan by acknowledging that contemporary sexual business has taken forms beyond ordinary imaginations (Tanaka 2010, pp. 9–19). He tries to locate the sexuality issue between pro-eros and con-eros. Nishimura's studies on the history of love dolls in Japan are among the few scholarly discussions of the subject (Nishimura 2008, 2017). Kikuchi (2018) offers an introduction to the studies of dolls media culture, dedicating part of his analysis to love dolls in Japan.

Sex dolls and sex robots are not exactly the same, but quite often, the latter are regarded as a more developed form of the former. There are obvious and apparent differences between the two of them, but this does not require us to draw any conclusions about the differences between a human–sex robot relationship and a human–sex doll relationship. Still, it is quite useful at this point to survey the contemporary condition of the Japanese love doll market.

Nowadays, there are several love doll sellers and rental service companies in Japan. The pioneering Japanese love (sex) doll company is Orient Industry, which produced their first doll in 1977. Hideo Tsuchiya, president of Orient Industry, explains how he came up with the idea of making a love doll. Back then, there were many different kinds of sex toys for women, but few options for men beside inflatable sex dolls. Therefore, he planned to design new models that were more physically attractive, with better structure and durability (Orient Industry 2017). Since then, there have been many developments, but it is unnecessary to trace all of them here. It is sufficient to note that current Orient Industry doll prices differ according to customers' choices and customization, varying between 480,000 and 700,000 yen. Orient Industry emphasizes the sexual and aesthetic splendidness of its sex dolls—among its staff, there are employees with art degrees. The customization options include body type, skin color, breast size, nipple color, head shape, the possibility of moving eyes and fingers, and the amount of pubic hair. Recently, Orient Industry began to make casts for female doll-bodies out of a real woman's cast.

Besides Orient Industry, the current Japanese sex doll market has a variety of stores and manufacturers, such as Harumidesigns (since 1984)—whose dolls' faces are more similar to Japanese women—4Woods (since 2002) and Project Level-D, as well as foreign doll retailers (mainly from China). Online stores such as Amazon and AliBaba also offer sex dolls for sale. Nowadays, most dolls are made out of silicone—which provides a harder, stiffer touch, and is sold at higher prices but also lasts longer—and TPE (thermoplastic elastomer material)—a cheaper option that provides a softer skin touch and possibly deteriorates in a shorter amount of time.

For those who still cannot afford, or would rather not buy, their own doll, there are options for rental, which are charged by the hour, and can be delivered to hotels or the customer's address, offering different types of looks and clothing. Companies such as Pretty Doll (since 2005) and Ero Ero Tenshi (since 2010) offer a delivery rental service, and

also have agreements and partnerships with hotels so they can prepare the room and the doll for customer use. Recently in the US, a debate took place over whether or not it would be acceptable to have a love doll brothel. In Japan, love doll rental has been available for almost fifteen years, but there have not been acknowledged public objections to it. This difference highlights the different cultural values and approaches to sexual culture represented by sex dolls.

Although there are a variety of Japanese sex doll manufacturers and retailers, not to mention a long and rich sexual cultural history and advanced robotics developments, there is not yet a national company that produces so-called sex robots, such as those being made in countries like China and the United States which aim to provide a more interactive experience for users. It is important to mention that beyond technological and material problems—such as possible transformations of the whole production process for developing and manufacturing sex robots in Japan—there might also be cultural, religious and spiritual concerns associated with the new technology. It is also interesting to note that new artistic and technological developments have explored and opened up new sexual cultures in Japan.

With the historical and social backgrounds of the sexual cultures of contemporary Japan, it would be possible to regard Mr. Kondo's case and the love doll owners' cases—human love for virtual and material figures—as a contemporary example of the human and non-human personal—and sometimes sexual—relations which the latest technology has made possible. Saying so does not suppose any "essential" Japanese culture behind Mr. Kondo's case, but is rather close to Sahlins' argument that out of the cultural structure, a historical event is formed and emerges (Sahlins 1987). From the etic perspective, the technological aspect receives the most attention, but from the emic perspective, his life story of the healing process with the virtual figure's help is his personal ground, out of which his practice of marriage with the virtual figure Hatsune Miku occurred. Furthermore, the technological side provides new and innovative conditions, in which Mr. Kondo's case is situated. The next section will deal with this issue.

## 5. Techno-Sexuality, Affection and Homeostasis

In this section, we will examine related issues such as the new conceptions and possibilities of sexuality in technologically enhanced scenarios, the soft robotics proposal (Damasio and Man 2019), and homeostasis to interpret the significance of the combined cases of Mr. Kondo, and love doll and Robohon owners in the current context. First, we will examine some of the currently available scholarly discussions, such as the concepts of posthuman sexuality as argued by Brett Lunceford (2009), Alexander Ornella (2009) and Michael Hauskeller (2014), techno-sexuality by Dennis Waskul (2014), and most recently, digisexuality by McArthur and Twist (2017).

The notion of posthuman sexuality has been examined from several perspectives. Discussing the body and the sacred in the digital age, Lunceford (2009) analyzes possible implications of posthuman existences, thinking specifically about the spiritual aspects of sexuality and what could be changed with new possibilities of sexual interaction through technological developments. He argues that "there seems to be something about sexuality that transcends culture" (Lunceford 2009, p. 82). The sexual transcends culture at the point at which it becomes spiritual. In other words, sexuality is closely intertwined with the notion of the sacred, as it remains a mystery among societies. From a different area of expertise, Lora Haddock, CEO of Lora DiCarlo—the company responsible for Osé, an award-winning innovative sex tech device at the 2019 CTA (Consumer Technology Association)—said in an interview that "we view female sexuality as sacred" (Wong 2020). She implies that, with sex tech, women learn to regard their sexuality as sacred without any male involvement. Though the article does not articulate her point of view much further, it is interesting and vital to recognize the spiritual and religious evaluation of sexuality assisted and enhanced by technological innovation.

Ornella ([2009](#)) analyzes how science-fiction narratives and posthuman philosophy imagine the transformation of sexual relationships with the development of new technological appliances. The notion of the posthuman is still new, fluid, and unstable, "suggesting that there is no common denominator on what exactly the posthuman being might look like and what his/her relation to today's human beings might be" (Ornella 2009, p. 314). According to Bradoitti ([2013](#)), "the posthuman condition introduces a qualitative shift in our thinking about what exactly is the basic unit of common reference for our species, our polity and our relationship to the other inhabitants of this planet" (Bradoitti 2013, p. 2). The author highlights that the posthuman circumstance would enforce the necessity to rethink the status of the human and its subjectivity and reinvent forms of ethical relations, norms, and values according to our times. Though scholars have widely used the notion of posthuman to describe a potential new condition that would emerge from a technologically enhanced scenario, it is important to note that there are different understandings of the subject (Gladden 2018), considering that the idea of posthuman would assume that there is an established and predetermined definition of what it means to be human. Authors like Clark ([2003](#)), for example, argue that we, as humans, are natural-born cyborgs, and that "what makes us distinctively human is our capacity to continually restructure and rebuild our own mental circuitry, courtesy of an empowering web of culture, education, technology, and artifacts" (Clark 2003, p. 10).

Hauskeller ([2014](#)) analyzes how human enhancement would change our sexual practices and how we relate emotionally, exploring not only mythological, historical, and literary predictions on the subject but also currently available and future sex/love-related products such as sex robots. When looking especially at sex robots, the author mentions that, as early as 1909, William James briefly mentioned the possibility of a sexual partner that could act just like a human lover would, but would not be able to feel anything at all, coining the term "automatic sweetheart". Exploring the available sex robot options, Hauskeller ([2014](#)) highlights that even though they are machines that do not think or feel, they are programmed to act as if they did. He also mentions that "it is interesting to see how the line between having feelings and reacting in a way that in humans we would see as a sign of feelings is deliberately blurred by the language that is being used in order to sell the robots to the customer" (Hauskeller 2014, p. 13). In this sense, Damasio and Man ([2019](#)) question what conditions would allow robots to care about what they do or think, proposing the construction of a new class of machines that would follow the principles of life regulation (homeostasis) and therefore, exhibit the equivalent of feelings.

Waskul ([2014](#)) proposes the term "techno-sexuality", referring to "the increasingly ubiquitous use of technology to gather sexual information, express sexual desires, view or expose sexual bodies, experience sexual pleasure, and explore sexual fantasies" (Waskul 2014, p. 94). He highlights the importance of thinking about sexuality from an individual point of view and from an institutional perspective, which creates patterns of normative behaviors within a specific culture.

McArthur and Twist ([2017](#)) also point out the rise of digital sexual cultures, coining the term "digisexuality" as a reference to sexual experiences that depend on advanced technology. According to them, there are several different stages of digisexuality. First-wave digisexuality would be based upon mediating technology enabling sexual connections between two persons. In contrast, second-wave digisexuality could be characterized as an immersive technology, depending not on human partners but on non-human partners such as robots, artificial agents, virtual and augmented reality. Digisexuals would consider technology-mediated sexual experiences as essential to their sexual identity. In this sense, Flore and Pienaar ([2020](#)) argue that intimacy and pleasure enhanced by teledildonic-enhanced sex can be called "sexuotechnical-assemblage", highlighting the uniquely technological dimension of sexuality.

While those authors pay more attention to the technological influence on human sexuality, Danaher ([2020](#)) cautions that the categorization of a new sexual identity label could lead to the pathologization and "othering" of "what should be viewed as part of

the ordinary range of human sexual desire" (Danaher 2020, p. 405). He refers to Ayala's conceptual act theory of sexual orientation, by which the latter means that humans have many different phenomenological experiences in their lifetimes and experience sexual desires, arousal and release in response to many different things.

Recently, when thinking about the concept of humanity and the development of artificial forms of life, Damasio and Man (2019) propose the construction of robots following the principles of homeostasis (life regulation), suggesting that, instead of characteristics of resilience, they would be equipped with some kind of vulnerability. According to Damasio (2018), the concept of homeostasis would relate to "the fundamental set of operations at the core of life, from the earliest and long-vanished point of its beginning in early biochemistry to the present . . . [a] powerful, unthought, unspoken imperative" (Damasio 2018, p. 27). In this sense, humans would distinguish themselves from other living beings due to their culture, understood as a set of different practices, objects, and ideas (such as art, philosophy, moral system, technology, and science, for example), and that feelings would be the motives and maintainers of these human accomplishments (Damasio 2018). By his definition, feelings would be connected to homeostasis: a deficient homeostatic process would be expressed by negative feelings, and an appropriate level process would originate from positive feelings. Damasio (2018) argues that cultural practices and the creation of different instruments started from a homeostatic decline or need (such as pain, suffering, a threat or loss).

Damasio (2018) also mentions that artificial intelligence programs that are being developed nowadays are still far from acquiring a condition similar to the human, since they do not develop feelings, which generate our vulnerabilities and everything that comes from them: suffering, joy or empathy. Even considering that there are options that reproduce human expressions through gesture, speech or facial movements, the author highlights that they do not function because of the robot's internal state, but are programmed by their creators. At the same time, people interact with, and can be enchanted by, robots in different ways. In this understanding, the construction of affection is built on the interaction between humans and robots.

Considering the cases of Mr. Kondo, and sex doll and Robohon owners, we could infer from Mr. Kondo's testimonials and part of the owners' responses that their partners had a healing effect on them. In moments of vulnerability—such as those described by Mr. Kondo and Mr. Nakajima, for example, on the emotional attachment they developed for their partners (Hatsune Miku and love dolls, respectively)—the latest technologies created a posthuman condition, helping them overcome difficult times and make a move toward life. Taking the Japanese context as an example and Damasio's (2018) finding that feelings constitute what singularizes the human species and guarantee the homeostatic process, we understand that, in vulnerable or precarious situations (such as loneliness or disease), robots (and other non-human entities) might grant the status/definition of a person for the humans to whom they relate. From the given examples, we could assume that the affective sphere has possibly taken an important role in this kind of development. Beyond the functional use or purpose, the feelings that emerge from these interactions might be key for exploring new features and improvements. From our examinations, Mr. Kondo's case could be better understood and appreciated in terms of possible new understandings of affection and digital intimacy instead of overemphasizing sexual aspects as existing studies do.

## 6. Conclusions

This paper attempted to explore an emic interpretation of Mr. Kondo's case—which the etic and Western studies have regarded as cases of digisexuality and AI sexuality—first by supplementing it with the authors' original studies on two other related issues of love doll owners and Robohon owners, and then by locating them both in the historical background of Japanese sexual and spiritual cultures, as well as in a context in which the human condition has been impacted by technological innovations at different levels. While

the etic and Western scholarly studies tend to regard Mr. Kondo's case as an example of digisexuality and AI sexuality, something emergent from the latest developed technology, this study shows that there are rather strong cultural continuities and cultural affinities in terms of human sexuality with non-human entities, as seen in Japanese sexual history examples such as *Shunga* paintings. Beyond sexual practices, the artificial partnership is better understood in terms of healing and affective correlation as a part of the homeostatic process.

Our original contribution to existing scholarship is our proposal to look at Mr. Kondo's case and similar ones in terms of affection and digital intimacy rather than focusing on sexuality, which is important but not necessarily and exclusively dominant in Japanese cases. Our research results also highlighted the belief of some love doll and Robohon owners that their respective partners had a heart or were alive, which might impact the way they see and relate to their dolls and robots. When analyzing the relation between dolls and religion, we have also mentioned the existence of funeral rituals for both dolls and robots. This aspect emphasizes the need to analyze the aforementioned beyond-human relationships from spiritual and religious perspectives too, considering possible significant differences in cultural understanding according to each group.

The limitation of this paper is conditioned by the difficulty in obtaining enough responses from love doll owners, as well as finding empiric information on current sex robot owners, since they are not broadly available to the public and current owners may not want to share their experiences due to their privacy. In addition, research conducted online may impact the responses as well as the profile of the participants.

**Author Contributions:** B.Y.A. and T.K. developed the conceptualization and design of the study. B.Y.A. collected the questionnaires data and analyzed it with input and assistance of T.K. Writing, review and editing were made by B.Y.A. and T.K. All authors have read and agreed to the published version of the manuscript.

**Funding:** This research was partially funded by the Brazilian Ministry of Education—CAPES (Coordination for the Improvement of Higher Education) Doctoral Sandwich Programme (PDSE) 2019–2020, and also partially funded by JSPS KAKENHI20K20491.

**Institutional Review Board Statement:** Ethical review and approval were waived for this study, since the collected information cannot readily identify the subjects and any disclosure of responses outside of the research would not reasonably place the subject at risk. Both questionnaires did not request name or contact information whatsoever.

**Informed Consent Statement:** Both the questionnaires applied in this article included an introduction statement and explained their purpose and intent of use. The questionnaire for love doll owners was shared on a public page on Twitter. The form for Robohon owners was shared in a closed Facebook group with the approval of the group administrators. All the questions included in the questionnaires applied were optional and should be responded only by the interested parts.

**Data Availability Statement:** Not applicable.

**Conflicts of Interest:** The authors declare no conflict of interest.

## Notes

1    According to a BBC article (Park 2020), skin hunger can be defined as the need to touch or be touched socially.
2    2020 might also be remembered as a year when sex dolls became a little bit more socially acceptable in the public space. The Campaign against Sex Robots (https://campaignagainstsexrobots.org/, accessed on 15 October 2020) opposes this social trend of accepting sex dolls in a public space. In the United States, for example, a restaurant created a different atmosphere by filling some of its seats with sex dolls and offering them as companions for a meal (Mencken 2020; Christian 2020a). In South Korea, the FC Seoul soccer team had sex dolls filling the stands during a game without spectators, which caused discomfort for some of the team's fans and became worldwide news (Sang-Hun 2020; Christian 2020b).
3    For example, the doll company Trottla is often referred to by many Western scholars discussing child sex dolls based upon an English interview essay in the American magazine, *The Atlantic* (Morin 2016). Other Japanese interviews with

the craftsman at Trottla show that he was misrepresented in this much-referenced essay. The article's author seems not to grasp Japanese sexual culture and history, due to their unfamiliarity with the Japanese language and culture.

4    This characterization poses a question to religious scholars because de Brosses' contribution is regarded as one of the origins of the modern study of religion.

5    It is important to mention that the authors of the present article have conducted a preliminary survey with Japanese doll owners, but there have been just 12 responses at the time of writing, an insufficient number to draw a general conclusion.

6    Hatsune Miku is a character originally created by company Crypton Future Media in 2007 as an illustration for Vocaloid, a singing voice synthesizer software. The character captivated users and soon became the vocalist of more than one hundred thousand songs (Zaborowski 2016), mostly produced by her fans, performed holographic live concerts and international tours and starred in movies, comics, and commercials.

7    The tale tells the story of Komurasaki, a young woman who falls in love with Gensuke, an employee at her parents' shop, who also falls in love with her. Their love story, however, is ruined by Komurasaki's parents, who marry her to another man. On their wedding night, Gensuke shows up as a ghost (*bakemono*, 化け物), shaped as a horned demon, haunting the newlywed husband. When he tells the story to his wife's parents, they have the idea of producing a doll that looks like their daughter, even including an artificial vagina (*azuma-gata*, 吾妻形). At night, the husband lies down with the doll and waits for the ghost in the bedroom, who reappears eager to kill his rival. However, when he sees the doll, Gensuke feels an uncontrollable sexual attraction towards her, takes her into his arms, forgets about the murder and never returns again.

8    The Mito Tokugawa family was a branch of the Tokugawa clan, a Japanese dynasty that was once a *daimyou* (feudal lords) family of Japan. The Mito Tokugawa collection contained dildos (*harigata*, 張型), accompanied by a note dated 1835, which described the characteristics and use for the pieces, as well as accessories for the penis, a ball for vaginal use, an artificial vagina (*gyokumongata* 玉門形, also popularly known as *azuma-gata-*吾妻形), and metal surgical tongs.

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
