# Peer review of "Sexuality and Affection in the Time of Technological Innovation: Artificial Partners in the Japanese Context"

_religions, doi:10.3390/rel12050296_

Round 1
Reviewer 1 Report
What an interesting article!
Adding a couple of images or graphs (e.g., result) will be helpful. It is my understanding, the Robot with a human figure is called Humanoid Robot.
FYI:
-Human-Robot Interaction by González-González et al. 2021 ( a very good review of recent articles).
-Sex Dolls by Knox et al. 2017? (Journal of Positive Sexuality)
Author Response
Dear reviewer,
Thank you for the careful reading of our manuscript and the article suggestions.
They were both great contributions and were included on the literature review (pages 3, 4 and 5).
As for the humanoid robot definition, we have complemented the information on the first paragraph of page 3.
As suggested, we have also inserted images and graphs which we believe that would contribute to the illustration of the given examples (pages 8, 11, 12, 13 and 14). Due to the graph insertions, we have made changes on the texts of pages 10, 11, 12 and 13 to facilitate the readers’ understanding of the data. We have also made a typographical correction on page 12.
We also highlight changes on the abstract (page 1), correcting a typographical error; as well as a correction of a reference on page 4, and a format change on page 6. On page 13, we have included some information on the subtopic 3.2.2, referring to the results of the questionnaire with Robohon owners.
On page 15, we have made an amendment pointed out by Reviewer 2.
On pages 24, 25 and 26, we have corrected a reference and included two new ones, as kindly recommended by Reviewer 1.
All changes were made using the “Track changes” function and highlighted in red.
Thank you again for your reading and contributions.
Kindest regards,
Beatriz Aoki and Takeshi Kimura
Reviewer 2 Report
The article contains 16,242 words, which is about 57 standardized pages. It has seven sections: 1. Introduction; 2. Literature Review; 3. Research Results: Three Contemporary Japanese Cases; 4. Human Figures, Sexual Culture and Spirituality in Japanese History; 5. Techno–Sexuality, Affection and Homeostasis; 6. Conclusions; and the list of References that contains 84 sources.
Authors seek a more inclusive perspective on the relationship between both religion and spirituality, and robots. Although an increase in scholarly debate in this area is present, it's culturally biased since it's focused either on Christianity or institutionalized religion. They chose the Japanese cultural history of sexuality due to its breadth.
In the introductory part, an extensive review of the literature is presented. Both historical and conceptual summary is provided. Cultural differences between the Western and Japanese understanding of sexuality are given. It ends with contemporary examples related to pleasure dolls (Japanese term). A case story of Mr. Akihiko Kondo who married virtual character Hatsune Miku is presented also.
A 2141-item questionnaire is issued to 12 Japanese respondents who were owners of pleasure dolls. The small sample size is explained due to shyness and reluctance to answer the questions about the pleasure dolls. A second questionnaire was distributed to Robohon owners; 39 responses were collected.
Results were presented as both quantitative and qualitative analyses. Due to the small sample size, no inferential statistics are present. They are followed by an overview of Japanese culture specifics due to both sexuality and the history of mechanical dolls usage in general.
Finally, some considerations about the future of sexuality are put in the theoretical frame of post-humanism. Perspectives from soft robotics, post-human sexuality, techno-sexuality, and digisexuality are considered.
I believe this paper stands as a good starting point for future both theoretical and empirical research in the field of post-humanistic sexuality. Some methodological limitations are present due to the small sample size, but I strongly believe that in the future, with intercultural perspectives included, we will be presented with larger and international samples.
Some corrections should be made, though:
[p12] „...transformations that took place after the Meiji modernization at the end of WWII and in post-WWII society.” >> Meiji Era took from 1868 to 1912
Author Response
Dear reviewer,
Thank you very much for the careful and detailed reading of our article.
We also see the research as a starting point and hope to deepen the discussions and overcome present limitations such as few empirical material and the questionnaire’s sample size.
We’d like to thank you also for your attention to the information on page 12 (now page 15) regarding the Meiji Restoration. We have made an amendment to correct this phrase.
Here we will point out all the other changes and inclusions made throughout the manuscript, according to the reviewers’ contributions:
We have made one change on the abstract (page 1), correcting a typographical error.
On page 3, we have complemented the information on the first paragraph as pointed out by Reviewer 1 and on pages 3, 4 and 5 we have cited two new articles suggestions. There was also the correction of a reference on page 4, and a format change on page 6.
Following the reviewer’s suggestion, we have also inserted images and graphs which we believe that would contribute to the illustration of the given examples (pages 8, 11, 12, 13 and 14). Due to the graph insertions, we have made changes on the texts of pages 10, 11, 12 and 13 to facilitate the readers’ understanding of the data. We have also made a typographical correction on page 12.
On page 13, we have included some information on the subtopic 3.2.2, referring to the results of the questionnaire with Robohon owners.
On pages 24, 25 and 26, we have corrected a reference and included two new ones, as kindly recommended by Reviewer 1.
All changes were made using the “Track changes” function and highlighted in red.
Thank you again for your reading and contributions.
Kindest regards,
Beatriz Aoki and Takeshi Kimura